# Grain Foods in US Infants Are Associated with Greater Nutrient Intakes, Improved Diet Quality and Increased Consumption of Recommended Food Groups

**DOI:** 10.3390/nu11122840

**Published:** 2019-11-20

**Authors:** Yanni Papanikolaou, Victor L. Fulgoni

**Affiliations:** 1Nutritional Strategies, 59 Marriott Place, Paris, ON N3L 0A3, Canada; 2Nutrition Impact, 9725 D Drive North, Battle Creek, MI 49014, USA; vic3rd@aol.com

**Keywords:** NHANES, infants, nutrients, grains, diet quality, food groups

## Abstract

There are limited data providing guidance on grain foods as part of a healthy dietary pattern in infants and may represent a gap in knowledge for the development of the 2020–2025 Dietary Guidelines for Americans Scientific Advisory Committee report currently in progress. An analysis using infant data from the National Health and Nutrition Examination Survey was conducted to assess grain food relationships with nutrient and energy intakes, diet quality, and food group consumption in infant consumers relative to non-consumers. Grain consumers were defined as infants consuming foods from the main grain food group, as defined by the US Department of Agriculture, and included whole and refined/enriched grains. All infants consuming grain foods had greater energy (kcal) vs. grain non-consumers (*p*’s < 0.0047). While infant grain consumers 6- to 12-months-old (*N* = 942) had higher daily intakes of sodium and added and total sugars, these infants also had significantly higher dietary fiber, calcium, folate, potassium, magnesium, zinc, phosphorus, choline, thiamin, riboflavin, and vitamin B6 compared to non-consumers. In 13- to 23-month-olds (*N* = 1668), grain consumption was associated with greater daily dietary fiber, iron, zinc, magnesium, phosphorus, folate, riboflavin, niacin, thiamin, vitamin A, vitamin B6, and vitamin B12 relative to non-consumers. Diet quality scores were significantly higher in all infant grain consumers examined in comparison to non-consumers (*p*’s < 0.0065). Grain intake was also linked with greater daily intake of several recommended food groups in both younger and older infants versus non-consumption of grains. The current analysis provides evidence to substantiate the inclusion of whole and enriched grain foods as part of the infant dietary pattern as beneficial associations between grain food consumption and dietary quality are apparent. Eliminating and/or reducing grain foods in infant dietary patterns may lead to unintended nutrient and health consequences.

## 1. Introduction

Current dietary guidance for Americans acknowledges the robust connection between improved nutrition and diet quality, regular physical activity and maintenance of good health and risk reduction of chronic diseases [1]. Historically, dietary guidance has provided evidence-based nutrition and food-based recommendations for Americans two years of age and older; thus, at present, there are no comprehensive dietary and nutrition recommendations for American children in the first two years of life [2]. In 2012–2013, a partnered initiative between the United States Department of Agriculture (USDA), the US Department of Health and Human Services, and the National Institutes of Health began highlighting gaps in nutrition research, with the objective of generating nutrition guidance and recommendations for infants ≤ 2 years-old, to help guide the development of the upcoming dietary guidance for Americans [3]. Though the 2020–2025 Dietary Guidelines Scientific Advisory Committee is currently convening to generate nutrition and dietary pattern recommendations for Americans from birth into adulthood and the elderly years, there are limited data providing guidance on how grain foods can be incorporated as part of an infant’s healthy dietary pattern.

An analysis of usual nutrient intakes using data from the Feeding Infants and Toddlers Study 2016 (FITS 2016) identified approximately 20% of infants 6- to 11.9-months-old who are at risk for iron inadequacy, while more than 50% of the same infants had inadequate intake of vitamin D and vitamin E. Only 3.2% and 2.5% of toddlers 12- to 23.9-months-old were above the adequate intake recommendation for dietary fiber and potassium, respectively, while many toddlers exceeded the upper limits for sodium, retinol and zinc [4]. While the FITS 2016 study provides data for overall dietary patterns, the contribution of grain food groups continues to be scarce. Data in older children (2–18 years-old) from the National Health and Nutrition Examination Survey (NHANES) 2005–2010 showed that daily dietary fiber intake was significantly higher in several grain food patterns of consumption, including grain foods predominantly from yeast breads, rolls, cereals, and pasta, in comparison to children not including grain foods in their diet. Additionally, iron intakes were significantly higher in all grain food patterns of consumption relative to children not consuming grains [5]. Similarly, national nutrition data on Canadian children from the Canadian Community Health Survey (CCHS-Nutrition 2015) showed that several grain food dietary patterns were associated with beneficial nutrient intakes, while children with limited grain foods in their diet (i.e., ≤1 serving of grains/day) had reduced intakes of fiber, magnesium, zinc, iron, calcium, folate, niacin, riboflavin, and thiamin [6]. Further research in children from the CCHS-Nutrition 2015 identified that intakes of calcium, iron and folate were greater when the diet was comprised of a balanced intake of whole and enriched grain foods, rather than only including one or the other, highlighting the value of a balanced approach to inclusion of grains in the diet and in achieving nutrient recommendations [7]. Recent US data from the NHANES has also shown that certain grains in preschool and school-aged children can contribute meaningful nutrient density in the diet, including greater percentages of dietary fiber, folate, iron, magnesium, zinc, thiamin, riboflavin, vitamin B6, vitamin B12, and vitamin A, relative to energy [8].

With limited evidence addressing nutrient needs and diet quality of infants in the scientific literature, particularly during the growth and development stages, analyses that consider the nutritional contribution of grain foods will help guide the evidence-based nutrition recommendations currently being developed by the Dietary Guidelines Scientific Advisory Committee. Therefore, relative to grain non-consumers, the present analysis aimed to determine grain food consumption in American infants and associations with nutrient and energy intakes, diet quality and food groups.

## 2. Experimental Section

The NHANES is a large database which provides a representative, cross-sectional survey of free-living American residents that is conducted every two years and from which data are released by the National Center for Health Statistics of the Centers for Disease Control and Prevention [9,10,11,12]. All required consents were previously obtained for all participants or proxies, and the protocol was approved by the Research Ethics Review Board at the National Center for Health Statistics. For the current analysis, data from eight NHANES datasets (2001–2002; 2003–2004; 2005–2006; 2007–2008; 2009–2010; 2011–2012; 2013–2014 and 2015–2016) were used, which provided data from children less than 2 years of age [13,14], resulting in a robust sample size (*N* = 3365). Nutrient intake data within the current analysis stem from the relevant USDA Food and Nutrient Database for Dietary Studies (FNDDS) [14,15]. Food group data were sourced from the USDA Food Patterns Equivalents Database [16]. The FNDDS contributes energy and nutrient values for all foods and beverages reported in What We Eat in America (WWEIA) [17], the dietary intake component of NHANES.

WWEIA is collected using the USDA’s Automated Multiple Pass Method (AMPM), a validated dietary data collection method which has proven to be an evidence-based, efficient and accurate format for collecting and assessing dietary intake data for national surveys [18]. AMPM is a fully computerized recall procedure that uses a five-step interview process [18]. Highly-trained interviewers with extensive knowledge of the food and nutrient intake files assess whether collected data is reliable or unreliable [18,19]. Previous data lends credibility to the AMPM model and has demonstrated the method’s accuracy with intake assessments [20]. While there are two dietary recall days in the NHANES methodology, only Day 1 offers in-person data collection [17,18]; hence, the present study only used Day 1 data in the analysis.

### 2.1. Subjects

The current NHANES analysis included all infants 6 to 23 months of age who were classified as either grain consumers or non-consumers (6–12 months: number [*N*] of consumers = 942; *N* of non-consumers = 574; 13–23 months: *N* of consumers = 1668; *N* of non-consumers = 181). Study participants were included in the analysis provided they demonstrated reliable and complete 24-hour dietary recalls. Grain consumers were defined as those infants consuming grain foods (with the exclusion of mixed dishes) and breastfeeding during the 24-hour dietary recall as defined by the USDA’s Food Patterns Equivalents Database [21]. 

### 2.2. Methods and Statistical Analyses

Statistical analyses were conducted using SAS software (Version 9.4, SAS Institute, Cary, NC, USA). Appropriate survey parameters were used to generate nationally-representative estimates for infants, which were also appropriately adjusted for the complex design of the NHANES. The USDA food coding system was used to identify and define grain foods. Specifically, as per the Food Patterns Equivalents Database, the grains category consists of two components: whole grains and refined grains. The USDA definition of whole grains encompasses amaranth, barley (not pearled), brown rice, buckwheat, bulgur, millets, oats, popcorn, quinoa, dark rye, triticale, wholegrain cornmeal, whole-grain wheat flour, whole-grain cracked wheat, wild rice, and grain-based products made with 100% whole grains or their flours. Additionally, refined grains are defined as grains that are degermed or polished and their flours or meal, cornmeal, masa, corn grits, bran of all cereals, cream of rice, cream of wheat, cracked wheat, malted barley or malted flours, pearled barley, rye (light and medium), wheat gluten, and white rice [21].

Least-square means and standard errors of daily energy (calories), nutrient intakes, food group consumption and diet quality were determined for infant grain consumers and non-consumers. Energy intake and diet quality components were adjusted for age, gender, ethnicity, and socioeconomic status (as assessed by the federally established criteria of the poverty income ratio [PIR]) [14]. Nutrient intakes were adjusted for age, gender, ethnicity and PIR, in addition to energy intake. Diet quality was assessed using the USDA’s Healthy Eating Index 2015 (HEI-2015), representing the most recent version of the HEI diet quality scale, and considered both the total and sub-component scores. HEI-2015 assesses diet quality and offers a quantitative assessment of conformance to the Dietary Guidelines for Americans, and serves as a tool to measure and track the dietary practices of Americans [22,23]. While the HEI has been created for Americans older than two years of age, most HEI measures are adjusted for energy and, therefore, it was determined that HEI represents an effective and appropriate tool for use in the present research design. 

## 3. Results

### 3.1. Energy and Nutrients

Table 1 (6- to 12-month-old infants) and Table 2 (13- to 23-month-old infants) represent adjusted energy and nutrient intakes for grain consumers and non-consumers. While grain consumption was associated with greater intake of energy, sodium and added and total sugars in infants 6- to 12-months of age, intakes of all nutrients were greater in grain consumers than in non-consumers, with the exception of intakes of iron, vitamin C, and vitamin E. Contrary to the younger infants, there were no differences in sodium and added and total sugars intake in grain consumers relative to non-consumers. Grain consumption in the older infants was associated with greater daily calories, dietary fiber, iron, zinc, magnesium, phosphorus, riboflavin, folate, thiamin, niacin, vitamin A, vitamin B6, and vitamin B12 relative to non-consumers of grains. 

### 3.2. Diet Quality: USDA Healthy Eating Index-2015 Total and Sub-Component Scores

Mean total and sub-component scores in infants for the HEI are displayed in Table 3 (6- to 12-month-old infants) and Table 4 (13- to 23-month-old infants). Younger grain consuming infants had significantly lower scores for sodium, but also had greater scores for greens and beans, total fruit, whole grains, refined grains, dairy foods, total protein foods, seafood and plant protein foods, and saturated fat in comparison to non-consumers of grains. Older infants consuming grains had greater scores for total fruit, whole fruit, whole grains, and refined grains relative to non-consumers. No differences were observed for protein foods or sodium scores between consumers and non-consumers. Grain consumers in both age groups had significantly higher total HEI scores relative to infant non-consumers.

### 3.3. Food Group Intake in Infants

Mean food group intake in infants for grain consumers and grain non-consumers is shown in Table 5 (6–12-month-old infants) and Table 6 (13–23-month-old infants). Infants 6–12 months-old had significantly higher intakes of milk, cheese, and total dairy foods compared to grain non-consumers. Infant grain consumption was linked to higher refined and whole grain intake, as well as greater intake in terms of total fruits, total vegetables and total meat, poultry, seafood, nuts and seeds compared to non-consumers of grains. 

No differences between grain consumers and non-consumers were seen for milk, yogurt, total dairy foods, total vegetables, and total meat, poultry, seafood, nuts and seeds in infants 13–23 months of age. However, grain consumers in this age group had higher daily intakes of cheese, refined and whole grains, and total fruits versus grain non-consumers.

## 4. Discussion

At present, there are limited data considering dietary patterns of infants, with even less evidence available pertaining to the role of grain foods within this population. Based on our review of the literature, this is the first study in infants using data from NHANES to examine grain consumption and associations with nutrient intake, diet quality and food group intake. Grain food consumption in 6- to 23-month-olds was found to be associated with higher nutrient intakes, better diet quality scores, and increased amounts of foods to encourage when compared to grain non-consumers. Specifically, grain consumption in 6- to 12-month-olds was linked with greater daily intake of numerous essential nutrients relative to non-consumers. Similarly, grain consumption in older infants was associated with greater daily intake of dietary fiber, folate, iron, magnesium, niacin, phosphorus, riboflavin, thiamin, vitamin A, vitamin B12, vitamin B6 and zinc compared to non-consumers. While grain consumption was associated with significantly greater intake of sodium and added and total sugars in infants 6- to 12-months-old, the 13- to 23-month-old infants showed no differences in sodium or added and total sugars intake when comparing grain consumers to non-consumers. Grain consumers of both infant age groups had significantly better diet quality, as reflected by the higher total HEI scores, relative to non-consumers. This may be attributed to the scores contributed from greater intake of greens and beans, total fruit, whole grains, dairy foods, total protein, seafood and plant protein foods in younger infants, and greater intake of total fruit, whole fruit and whole grains in the older infants, when compared to the grain non-consumers. When assessing food group intake, younger infants tended to eat greater amounts of dairy foods, whole grains, refined grains, total fruits, total vegetables and protein-rich foods, while older infants had greater daily intake of cheese, refined grains, whole grains and total fruit, relative to non-consumers.

While our previous work did not consider the contribution of grain foods in infants, grain consumption in young children appeared to be beneficial, with grain foods acting as part of a healthy dietary pattern. Indeed, several types of grain foods, including refined and whole grains, contributed to nutrient density in US children aged 1- to 3-years. Specifically, grain foods accounted for approximately 13% of all calories, 16% of sodium, 7% of total sugar and 5% of saturated fat in the total diet, while providing about 23% of all dietary fiber, 40% of folate, 38% of iron, 30% of thiamin, 29% of niacin, 23% of vitamin B6, 19% of zinc, 18% of vitamin A, 18% of riboflavin, 17% of vitamin B12, and 13% of magnesium. Subcategories of grain foods, including breads, rolls, tortillas, and ready-to-eat cereals also contributed meaningful levels of nutrients, demonstrating that certain grain foods contribute nutrient density that surpasses caloric contributions in the diet [5]. Similarly, recent work using data from the NHANES 2015-2016 found that ready-to-eat cereal consumption in children aged 6 months to 17 years was associated with greater daily intake of carbohydrates, total sugar, dietary fiber, calcium, zinc, magnesium, iron, potassium, folate, niacin, riboflavin, thiamin, vitamin A, vitamin B6, vitamin B12, and vitamin D compared to non-consumers of ready-to-eat cereals. Dietary folate, thiamin, and vitamin B6 daily intakes were significantly greater in infant ready-to-eat cereal consumers relative to non-consumers [24]. Similar to the current findings, children aged 6-months to 17-years-old consuming ready-to-eat cereals showed significantly elevated intake of food groups to encourage, including 29% greater total dairy intake and 61% higher whole grain intake compared to non-consumers. Intake of milk and whole grains were significantly higher in infants and toddlers consuming ready-to-eat cereals when compared to the infant and toddler non-consumers [24], suggesting that grain foods may be a delivery vehicle for nutrient-rich dairy foods.

While the NHANES analysis in infant and toddler ready-to-eat cereal consumers did not find significant differences in energy intake [24], the current analysis showed that energy intake in grain consumers was consistently higher relative to non-consumers. Likewise, while the infant and toddler study showed no differences in sodium intake between ready-to-eat cereal consumers and non-consumers, the current analysis showed higher sodium intakes in grain consumers 6- to 12-months-old relative to non-consumers. This has also been documented in previously published work in preschool and school-aged children in the US and Canada [5,6]. Therefore, while grain foods can be contributors of energy and sodium to the diet of children, early acceptance and familiarity with nutrient-dense grain foods will likely help close nutrient intake recommendation gaps as children grow and develop into adulthood, with particular emphasis on Dietary Guidelines for Americans shortfall nutrients and nutrients of public health concern, which include, but are not limited to, dietary fiber, folate, magnesium, calcium, and iron [1]. 

The current study has limitations inherent in cross-sectional and/or observational research which have previously been reported. Importantly, the present results are observations between grain consumers, nutrient intakes, diet quality and food group consumption. Thus, the current observational research cannot be used to establish cause and effect. As the current findings are reliant on self-reported methods, this can potentially lead to under- or over-estimation of food intake, which can contribute inaccuracies in energy and nutrient intakes. Dietary recalls in infant research also rely on the memory of the parents or caregivers, and while validated procedures are established to gather nutrition data, bias from memory recall challenges may be introduced into the analysis [25]. Also, the current study evaluated dietary patterns with and without grain foods. Thus, other foods comprising the overall infant dietary pattern are likely to contribute to the observed associations. A substantial advantage of using the NHANES data includes a robust nationally-representative sample of American children with corresponding data on energy and nutrient intakes, as well as food and beverage consumption.

## 5. Conclusions

This is the first study to demonstrate differences in intakes of nutrients, diet quality and food group consumption between grain consumers and non-consumers in US infants. Grain consumption, in general, was associated with higher nutrient intakes, better diet quality scores, and increased amounts of foods to encourage when compared to grain non-consumers. Infants 6- to 12-month-olds had higher daily intake of sodium and added and total sugar, but also had significantly higher daily intake of dietary fiber, calcium, folate, magnesium, phosphorus, potassium, riboflavin, thiamin, choline, zinc and vitamin B6 relative to non-consumers. In addition, 13- to 23-month-old participants had higher daily intake of dietary fiber, folate, iron, magnesium, niacin, phosphorus, riboflavin, thiamin, vitamin A, vitamin B12, vitamin B6 and zinc. An investigation of food group intake showed that younger grain-consuming infants tended to eat greater amounts of several recommended food groups, including higher amounts of dairy foods, whole grains, total fruits, total vegetables and protein-rich foods, while older grain-consuming infants had greater daily intake of cheese, whole grains and total fruit compared to grain non-consumers. The current evidence supports the inclusion of grain foods in infant dietary patterns. Furthermore, eliminating or reducing grain foods in the diets of American infants 6–23 months-old may have unintended nutrient, food group and diet quality consequences in the future. While staying within caloric needs and recommendations, and being sensitive to added sugars, saturated fat, and sodium intake, caregivers are encouraged to select whole and enriched grain foods that contribute nutrient density.

## Figures and Tables

**Table 1 nutrients-11-02840-t001:** Daily mean energy and nutrient intakes in 6- to 12-month-old grain consumers vs. grain non-consumers.

	GRAIN NON-CONSUMERS		GRAIN CONSUMERS				
Energy/Nutrients	LSMean	SE	LSMean	SE	Beta	SE	*p*
Energy (kcal)	846	16	1047	22	201	28	<0.0001
Carbohydrate (g)	112	2	138	3	25	4	<0.0001
Added sugars (tsp eq)	1.0	0.15	2.8	0.17	1.8	0.22	<0.0001
Total sugars (g)	80	1.3	87	1.6	7	2.1	0.0029
Protein (g)	21	0.7	33	1	12	1.3	<0.0001
Total fat (g)	35	0.8	42	1.0	7	1.2	<0.0001
Monounsaturated fatty acids (g)	11.6	0.3	14.0	0.4	2.6	0.5	<0.0001
Polyunsaturated fatty acids (g)	7.2	0.2	7.7	0.2	0.5	0.3	0.0417
Saturated fatty acids (g)	14.5	0.4	16.8	0.4	2.3	0.6	0.0001
Dietary fiber (g)	4.5	0.2	6.6	0.2	2.2	0.3	<0.0001
Calcium (mg)	720	18	856	22	136	27	<0.0001
Folate, DFE (mcg)	193	4.6	270	8.7	77	10	<0.0001
Iron (mg)	17.9	0.6	15.4	0.4	−2.5	0.7	0.0008
Magnesium (mg)	109	2.8	140	3.3	31	4.3	<0.0001
Niacin (mg)	11.8	0.4	12.5	0.3	0.8	0.5	0.1202
Phosphorus (mg)	515	15	730	20	215	25	<0.0001
Potassium (mg)	1188	28	1521	33	333	44	<0.0001
Riboflavin (Vitamin B2) (mg)	1.4	0.03	1.6	0.04	0.2	0.05	<0.0001
Sodium (mg)	454	27	1006	44	552	50	<0.0001
Thiamin (Vitamin B1) (mg)	1.0	0.03	1.1	0.03	0.1	0.04	0.0047
Total choline (mg)	131	5.4	165	5.2	34	7.4	<0.0001
Vitamin A, RAE (mcg)	717	16	679	12	38	19	0.0458
Vitamin B12 (mcg)	2.2	0.07	3.1	0.1	0.9	0.1	<0.0001
Vitamin B6 (mg)	0.7	0.02	0.9	0.02	0.2	0.03	<0.0001
Vitamin C (mg)	106	3.2	100	3.2	−6.0	4.5	0.2292
Vitamin D (D2 + D3) (mcg)	9.06	0.2	9.08	0.2	0.02	0.2	0.9410
Vitamin E as alpha-tocopherol (mg)	8.4	0.2	6.9	0.2	−1.5	0.3	<0.0001
Zinc (mg)	6.6	0.2	7.3	0.2	0.7	0.2	0.0037

LSMean = least square mean; SE = standard error; Beta = regression coefficient for difference among groups; Data represent grain consumers (number [*N*] = 942) and grain non-consumers (*N* = 574); infants 6- to 12-months-old, gender combined; National Health and Nutrition Examination Survey [NHANES] 2001–2016; Day 1 intakes.

**Table 2 nutrients-11-02840-t002:** Daily mean energy and nutrient intakes in 13- to 23-month-old grain consumers vs. grain non-consumers.

	GRAIN NON-CONSUMERS		GRAIN CONSUMERS				
Energy/Nutrients	LSMean	SE	LSMean	SE	Beta	SE	*p*
Energy (kcal)	1166	45	1309	17	143	50	0.0046
Carbohydrate (g)	146	6	173	2	27	7	0.0001
Added sugars (tsp eq)	7.3	0.7	7.6	0.2	0.3	0.7	0.6423
Total sugars (g)	94	4.1	99	1.6	5	4.1	0.2500
Protein (g)	44	2	49	0.7	5	2.2	0.0188
Total fat (g)	46	2	48	1	2	2.5	0.3701
Monounsaturated fatty acids (g)	15.9	0.8	16.0	0.3	0.1	0.9	0.8819
Polyunsaturated fatty acids (g)	7.4	0.4	8.2	0.2	0.8	0.5	0.1074
Total saturated fatty acids (g)	18.8	0.9	19.7	0.4	0.9	1.0	0.3475
Dietary fiber (g)	6.0	0.4	8.9	0.2	2.9	0.4	<0.0001
Calcium (mg)	962	50	1042	18	80	53	0.1321
Folate, DFE (mcg)	198	12	335	6.7	136	14	<0.0001
Iron (mg)	8.1	0.6	9.7	0.2	1.6	0.7	0.0197
Magnesium (mg)	156	7.5	181	2.3	25	8.0	0.0024
Niacin (mg)	9.4	0.5	12.2	0.2	2.8	0.5	<0.0001
Phosphorus (mg)	935	44	1043	15	108	48	0.0256
Potassium (mg)	1838	85	1983	27	145	92	0.1178
Riboflavin (Vitamin B2) (mg)	1.7	0.08	2.0	0.03	0.3	0.09	0.0105
Sodium (mg)	1594	145	1783	27	190	149	0.2063
Thiamin (Vitamin B1) (mg)	1.0	0.05	1.1	0.02	0.1	0.05	0.0002
Total choline (mg)	211	12.8	204	3.4	−7.0	14.1	0.6318
Vitamin A, RAE (mcg)	455	27	564	12	109	30	0.0004
Vitamin B12 (mcg)	3.7	0.2	4.3	0.1	0.6	0.2	0.0107
Vitamin B6 (mg)	0.9	0.05	1.2	0.02	0.3	0.05	<0.0001
Vitamin C (mg)	81	9.2	83	2.5	2.0	9.0	0.8149
Vitamin D (D2 + D3) (mcg)	8.6	0.5	8.4	0.2	0.2	0.6	0.7625
Vitamin E as alpha-tocopherol (mg)	4.0	0.3	3.9	0.1	0.1	0.4	0.7889
Zinc (mg)	6.3	0.3	7.5	0.1	1.2	0.3	0.0005

LSMean = least square mean; SE = standard error; Beta = regression coefficient for difference among groups; Data represent grain consumers (*N* = 1,668) and grain non-consumers (*N* = 181); infants 13- to 23-months-old, gender combined; NHANES 2001–2016; Day 1 intakes.

**Table 3 nutrients-11-02840-t003:** Mean (standard error [SE]) total Healthy Eating Index-2015 and sub-component scores for 6- to 12-month-old grain consumers vs. grain non-consumers.

	GRAIN NON-CONSUMERS		GRAIN CONSUMERS		
HEI Scores	LSMean	SE	LSMean	SE	*p*
Component 1: Total vegetables	2.08	0.1	2.09	0.1	0.9768
Component 2: Greens and beans	0.24	0.1	0.61	0.1	<0.0001
Component 3: Total fruit	3.24	0.1	3.60	0.1	0.0079
Component 4: Whole fruit	3.11	0.1	3.31	0.1	0.1916
Component 5: Whole grains	1.91	0.2	2.63	0.1	0.0012
Component 6: Dairy	1.71	0.2	4.45	0.2	<0.0001
Component 7: Total protein foods	0.90	0.1	1.88	0.1	<0.0001
Component 8: Seafood and plant protein	0.23	0.1	0.65	0.1	<0.0001
Component 9: Fatty acid ratio	2.02	0.1	1.96	0.1	0.6418
Component 10: Sodium	9.66	0.1	8.72	0.1	<0.0001
Component 11: Refined grains	9.62	0.1	8.90	0.1	<0.0001
Component 12: Saturated fat	2.30	0.2	2.92	0.2	0.0033
HEI-2015 TOTAL SCORE	46.73	0.5	51.2	0.4	<0.0001

LSMean = least square mean; SE = standard error; Beta = regression coefficient for difference among groups; Data represent grain consumers (*N* = 942) and grain non-consumers (*N* = 574); infants aged 6- to 12-months-old, gender combined; NHANES 2001–2016; Day 1 intakes.

**Table 4 nutrients-11-02840-t004:** Mean (SE) total and sub-component scores for the Healthy Eating Index 2015 (HEI-2015) in 13- to 23-month-old grain consumers vs. grain non-consumers.

	GRAIN NON-CONSUMERS		GRAIN CONSUMERS		
HEI Scores	LSMean	SE	LSMean	SE	*p*
Component 1: Total vegetables	1.97	0.2	2.01	0.1	0.8504
Component 2: Greens and beans	0.99	0.2	0.87	0.1	0.5718
Component 3: Total fruit	3.40	0.2	3.94	0.1	0.0143
Component 4: Whole fruit	2.68	0.2	3.30	0.1	0.0143
Component 5: Whole grains	1.00	0.2	2.54	0.1	0.0090
Component 6: Dairy	8.47	0.4	8.98	0.1	0.2296
Component 7: Total protein foods	3.19	0.2	3.00	0.1	0.2955
Component 8: Seafood and plant protein	1.36	0.2	1.26	0.1	0.6996
Component 9: Fatty acid ratio	1.89	0.3	1.69	0.1	0.5499
Component 10: Sodium	6.76	0.3	6.78	0.1	0.9672
Component 11: Refined grains	8.56	0.3	7.56	0.1	0.0030
Component 12: Saturated fat	2.97	0.4	3.71	0.1	0.0539
HEI-2015 TOTAL SCORE	51.00	1.0	53.76	0.4	0.0064

LSMean = least square mean; SE = standard error; Data represent grain consumers (*N* = 1,668) and grain non-consumers (*N* = 181); infants aged 13- to 23-months-old, gender combined; NHANES 2001–2016; Day 1 intakes; Covariates include age, gender, ethnicity, and poverty income ratio.

**Table 5 nutrients-11-02840-t005:** Mean (SE) food group intake for 6- to 12-month-old grain consumers vs. grain non-consumers.

	GRAIN NON-CONSUMERS		GRAIN CONSUMERS		
Food Group	LSMean	SE	LSMean	SE	*p*
Cheese (cup eq)	0.04	0.01	0.16	0.02	<0.0001
Milk (cup eq)	0.29	0.05	1.06	0.08	<0.0001
Yogurt (cup eq)	0.03	0.01	0.05	0.01	0.2249
Total Dairy (cup eq)	0.37	0.06	1.27	0.09	<0.0001
Refined Grains (oz eq)	0.71	0.05	1.67	0.10	<0.0001
Whole Grains (oz eq)	0.26	0.02	0.42	0.02	<0.0001
Total Fruits (cup eq)	0.67	0.03	0.99	0.04	<0.0001
Total Vegetables * (cup eq)	0.42	0.02	0.48	0.02	0.0475
Total Meat, Poultry, Seafood, Nuts & Seeds (oz eq)	0.43	0.05	1.10	0.07	<0.0001

LSMean = least square mean; SE = standard error; Data represent grain consumers (*N* = 942) and grain non-consumers (*N* = 574); infants aged 6- to 12-months-old, gender combined; NHANES 2001–2016; Day 1 intakes. * excludes legumes.

**Table 6 nutrients-11-02840-t006:** Mean (SE) food group intake for 13- to 23-month-old grain consumers vs. grain non-consumers.

	GRAIN NON-CONSUMERS		GRAIN CONSUMERS		
Food Group	LSMean	SE	LSMean	SE	*p*
Cheese (cup eq)	0.26	0.04	0.39	0.02	0.0011
Milk (cup eq)	2.10	0.17	2.25	0.05	0.3979
Yogurt (cup eq)	0.07	0.02	0.09	0.01	0.3480
Total Dairy (cup eq)	2.46	0.18	2.74	0.06	0.1258
Refined Grains (oz eq)	2.06	0.29	2.90	0.06	0.0053
Whole Grains (oz eq)	0.19	0.04	0.51	0.02	<0.0001
Total Fruits (cup eq)	1.15	0.11	1.41	0.04	0.0349
Total Vegetables * (cup eq)	0.49	0.05	0.55	0.02	0.2316
Total Meat, Poultry, Seafood, Nuts & Seeds (oz eq)	2.13	0.18	2.09	0.05	0.8134

LSMean = least square mean; SE = standard error; Data represent grain consumers (*N* = 1668) and grain non-consumers (*N* = 181); infants aged 13- to 23-months-old, gender combined; NHANES 2001–2016; Day 1 intakes. * excludes legumes.

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
