# Peer review of "Grain Foods in US Infants Are Associated with Greater Nutrient Intakes, Improved Diet Quality and Increased Consumption of Recommended Food Groups"

_nutrients, 2019, doi:10.3390/nu11122840_

Round 1

Reviewer 1 Report

This article titled "Grain Foods in Infants are Associated with Improved Nutrient Intakes, Better Diet Quality and Greater Consumption of Food Groups to Encourage" provides interesting associations between intake of grains and improvement in both quantity and quality of dietary consumption in infants/toddlers.

This study has some strengths (despite the limitations as pointed out by the authors) such as good sample size, and enriches our dietary understanding between grain consumers/non consumers in the included study age.

Minor comment:

I would recommend authors to give a better title (please consider to a better wording for the phrase - ".....food groups to encourage"

Author Response

Dear Reviewer,

Thank you for your comments and feedback. We greatly appreciate your recommendation to revise the title and as such, we propose editing the title to read:

"Grain Foods in Infant Dietary Patterns are Associated with Greater Nutrient Intakes, Improved Diet Quality, and Increased Consumption of Recommended Food Groups"

Please reach back with any questions or further comments.

Yanni Papanikolaou and Victor L. Fulgoni

Reviewer 2 Report

Explain somewhere why you chose grain consumption as the criterion.

The striking association between grain consumption and higher dietary quality - is it explained by the nutrient content of grain foods, or is the consumption of grain foods a marker of higher dietary quality? Do the authors have an opinion?

Line 13: change to "an analysis of NHANES data was conducted" Line 16: what is the "main grain food group" ? Line 29: change to "between grain food consumption and dietary quality" Line 36: change "developed" to "provided" Line 40: insert "the" before "US" Line 41: change "set forward" to "began" Line 43: change "-old, of which" to ",which" Line 49: change to "an analysis of usual nutrient intakes" Line 50: change "-months-old are" to "months old who are" Line 58: change "grains" to "grain' Line 62: change "reported" to "showed" Line 80: change "is a" to "resulting from a" Line 81: change "residents, where data are released every two years" to "residents, which is conducted every two years and from which data are released" Line 86: change "using" to "were used which provided data from infants ..." Line 100:change "methods" to "method's" Line 104: insert "who" after "age" Lines 108-109: what is the meaning of "with the exclusion of breastfeeding" Line 135: change "adjusted" to "adjusted for" Line 146: delete "in infants" Lines 147-150: change to "In infants 6 to 12 months of age, intakes of all nutrients were greater in grain consumers than in con-consumers with the exception of intakes of iron, Vitamin c and Vitamin E." Similar changes for text pertaining to Table 2 Line 191: change Table 3 to Table 5 and Table 4 to Table 6 Lines 219 to 224: do not repeat all the individual nutrients here Lines 269-270 change to "the current observations cannot be used to establish cause and effect" Tables: take away "Adjusted" from the legends of all tables

Author Response

Dear Reviewer,

Thank you for taking the time to provide feedback and recommended revisions and/or edits. Below are answers and comments to your questions/recommendations. Please let us know if you have further questions.

Sincerely,

Yanni Papanikolaou and Victor L. Fulgoni

Reviewer Feedback

Reviewer: Explain somewhere why you chose grain consumption as the criterion.

Authors’ Response: This has been included in the introduction section (refer to lines 74-77 of the attached revised manuscript). Please let us know if you feel further clarification is required. Thank you.

Reviewer: The striking association between grain consumption and higher dietary quality - is it explained by the nutrient content of grain foods, or is the consumption of grain foods a marker of higher dietary quality? Do the authors have an opinion?

Authors’ Response: This is a great question. Diet quality is measured by the Healthy Eating Index (HEI), and as such, the sub-categories of the HEI are weighted based on authoritative recommendations. Grains consumption and associations to a higher diet quality are likely attributable to a combination of factors. In the 6-12 month-old infants, we saw greater consumption of foods that typically compliment grains foods, including nutrient-rich dairy foods, total fruits, greens and beans, whole grains, and protein-rich foods. In the older infants, we saw greater consumption of total fruits, whole fruits, and whole grains, all of which are important contributors to the HEI scale. Thus, our current data shows the importance of the overall dietary pattern—one which includes whole and enriched grain foods. Please let us know if you have further questions here. Thank you.

Reviewer: Line 13: change to "an analysis of NHANES data was conducted"

Authors’ Response: This has been revised as per your recommendation. The sentence reads as follows: “An analysis using data from the National Health and Nutrition Examination Survey was conducted to assess grain food associations with energy and nutrient intakes, diet quality and food group consumption in infant consumers relative to non-consumers.”

We used ‘National Health and Nutrition Examination Survey to help better orient the reader. Thank you.

Reviewer: Line 16: what is the "main grain food group" ?

Authors’ Response: Main grain food group refers to the USDA categorization of grains which describes all grain foods combined. A subgroup grain food group would include items like breads, rolls, tortillas, ready-to-eat cereals, etc. This term is meant to distinguish grains in the grain group from grains in mixed dishes. Please let us know if you have further questions. Thank you.

Reviewer: Line 29: change to "between grain food consumption and dietary quality"

Authors’ Response: This has been edited as per your recommendation. Thank you.

Reviewer: Line 36: change "developed" to "provided"

Authors’ Response: This has been revised as per your feedback. Thank you.

Reviewer: Line 40: insert "the" before "US"

Authors’ Response: Thank you for highlighting this detail. This edit has been completed.

Reviewer: Line 41: change "set forward" to "began"

Authors’ Response: This has been edited as per your recommendation. Thank you.

Reviewer: Line 43: change "-old, of which" to ",which"

Authors’ Response: This has been edited as per your recommendation. Thank you.

Reviewer: Line 49: change to "an analysis of usual nutrient intakes"

Authors’ Response: This has been revised as per your feedback. Thank you.

Reviewer: Line 50: change "-months-old are" to "months old who are"

Authors’ Response: This edit has been completed. Thank you.

Reviewer: Line 58: change "grains" to "grain'

Authors’ Response: This change has been incorporated. Thank you.

Reviewer: Line 62: change "reported" to "showed"

Authors’ Response: This has been revised as per your feedback. Thank you.

Reviewer: Line 80: change "is a" to "resulting from a"

Authors’ Response: We would prefer to keep the sentence as is to align with USDA and CDC descriptions. The sentence reads, “The NHANES database is a nationally-representative, cross-sectional survey of non-institutionalized, civilian residents…”. Please let us know if you have further questions. Thank you.

Reviewer: Line 81: change "residents, where data are released every two years" to "residents, which is conducted every two years and from which data are released"

Authors’ Response: This has been revised as per your feedback. Thank you.

Reviewer: Line 86: change "using" to "were used which provided data from infants ..."

Authors’ Response: This has been revised as per your feedback. Thank you.

Reviewer: Line 100:change "methods" to "method's"

Authors’ Response: This has been revised. Thank you.

Line 104: insert "who" after "age"

Authors’ Response: Thank you. This has been edited accordingly.

Reviewer: Lines 108-109: what is the meaning of "with the exclusion of breastfeeding"

Authors’ Response: The analysis did not include infants who were breastfed in the analysis. Let us know if you have further questions. Thank you.

Reviewer: Line 135: change "adjusted" to "adjusted for"

Authors’ Response: Thank you for catching this detail. We have added the word ‘for’ as suggested.

Reviewer: Line 146: delete "in infants"

Authors’ Response: Thank you. We have deleted ‘in infants’ as recommended.

Reviewer: Lines 147-150: change to "In infants 6 to 12 months of age, intakes of all nutrients were greater in grain consumers than in non-consumers with the exception of intakes of iron, Vitamin c and Vitamin E." Similar changes for text pertaining to Table 2

Authors’ Response: This has been revised accordingly as per your feedback. Thank you. We kept the text unchanged for table 2 to provide a different style approach to describing the tables. Please let us know if you have further comments.

Reviewer: Line 191: change Table 3 to Table 5 and Table 4 to Table 6

Authors’ Response: Thank you. The edits have been completed.

Reviewer: Lines 219 to 224: do not repeat all the individual nutrients here

Authors’ Response: Thank you. This has been revised to read “…numerous essential nutrients”. Great observation!

Reviewer: Lines 269-270 change to "the current observations cannot be used to establish cause and effect"

Authors’ Response: The edits have been completed as recommended. Thanks.

Reviewer: Tables: take away "Adjusted" from the legends of all tables

Authors’ Response: The word ‘adjusted’ has been removed from all legends in the tables. Thank you.

Reviewer 3 Report

The manuscript by Papanikolaou and Fulgoni aimed to examine grain food consumption in infants in associations with nutrient intakes, diet quality and food groups. This was the first study to demonstrate differences in nutrient intakes, diet quality and food group consumption between grain consumers and non-consumers in US infants 6- to 12-months-old and 13- to 23-months-old. Grain consumption, in general, resulted associated with higher nutrient intakes, better diet quality scores, and increased amounts of foods to encourage when compared to grain non-consumers. However, I suggest to limit the results to US population changing the title.

Author Response

Dear Reviewer,

Thank you for your feedback. We agree with your recommendation to revise the title to specify US infants. As such, the newly proposed title will read:

"Grain Foods in US Infants are Associated with Greater Nutrient Intakes, Improved Diet Quality and Increased Consumption of Recommended Food Groups"

Please let us know if you have further questions and/or comments.

Thank you,

Yanni Papanikolaou and Victor L. Fulgoni

Round 2

Reviewer 2 Report

Line 15: delete "data"